# JHTDB-wind: a web-accessible large-eddy simulation database of a wind farm with virtual sensor querying

Xiaowei Zhu<sup>1</sup>, Shuolin Xiao<sup>2</sup>, Ghanesh Narasimhan<sup>3</sup>, Luis A. Martinez-Tossas<sup>4</sup>, Michael Schnaubelt<sup>5</sup>, Gerard Lemson<sup>5</sup>, Hanxun Yao<sup>6</sup>, Alexander S. Szalay<sup>7</sup>, Dennice F. Gayme<sup>8</sup>, and Charles Meneveau<sup>8</sup>

**Correspondence:** Charles Meneveau (meneveau@jhu.edu)

Abstract. This manuscript introduces JHTDB-wind (https://turbulence.idies.jhu.edu/datasets/windfarms), a publicly accessible database containing large-eddy simulation (LES) data from wind farms. Building on the framework of the Johns Hopkins Turbulence Database (JHTDB), which hosts direct numerical simulation and some large-eddy simulation datasets of canonical turbulent flows, JHTDB-wind stores the 4D space-time history of the flow and provides users the ability to access and query the data via a web-based virtual sensor interface. The initial dataset comprises LES results from a large wind farm with 6 × 10 turbines, modeled using a filtered actuator line method, under conventionally neutral atmospheric conditions. This data comprises one hour of flow field data (velocity, pressure, potential temperature deviation, subgrid-scale eddy viscosity, and turbine forces, approximately 15 TB) and wind turbine data—including both turbine-level operational quantities and blade-level aerodynamic quantities (approximately 1.3 TB)—stored in Zarr and Parquet formats, respectively. Data retrieval is facilitated by the Giverny Python package, allowing remote users to query the database in Python or Matlab (C and Fortran support are available for flow field data). This paper details the simulation setup and demonstrates data access through examples that analyze wind farm flow structures and turbine performance. The framework is extensible to future datasets, including the JHTDB-wind diurnal cycle simulation analyzed in Xiao et al. (2025).

# 1 Introduction

Eddy-resolving simulations of atmospheric boundary layer (ABL) phenomena (Porté-Agel et al., 2000; Bou-Zeid et al., 2004; Kumar et al., 2006) and of wind farms in particular (Calaf et al., 2010; Meyers and Meneveau, 2012; Gebraad et al., 2016; Stevens and Meneveau, 2017; Zhang et al., 2023) have significantly advanced our understanding of the complex, multi-scale,

<sup>&</sup>lt;sup>1</sup>Department of Mechanical and Materials Engineering, Portland State University, Portland, OR 97201, USA.

<sup>&</sup>lt;sup>2</sup>Ralph O'Connor Sustainable Energy Institute, Johns Hopkins University, Baltimore, MD 21218, USA.

<sup>&</sup>lt;sup>3</sup>Department of Mechanical Engineering and St. Anthony Falls Lab, University of Minnesota, Minneapolis, MN 55455, USA.

<sup>&</sup>lt;sup>4</sup>National Renewable Energy Laboratory, Golden, CO 80401, USA.

<sup>&</sup>lt;sup>5</sup>Institute for Data Intensive Engineering and Science, Johns Hopkins University, Baltimore, MD 21218, USA.

<sup>&</sup>lt;sup>6</sup>Department of Mechanical Engineering and Institute for Data Intensive Engineering and Science, Johns Hopkins University, Baltimore, MD 21218, USA.

<sup>&</sup>lt;sup>7</sup>Department of Physics and Astronomy, Department of Computer Science, and Institute for Data Intensive Engineering and Science, Johns Hopkins University, Baltimore, MD 21218, USA.

<sup>&</sup>lt;sup>8</sup>Department of Mechanical Engineering, Institute for Data Intensive Engineering and Science, and Ralph O'Connor Sustainable Energy Institute, Johns Hopkins University, Baltimore, MD 21218, USA.

and multi-physics processes involved. Large Eddy Simulations (LES) offer high spatial and temporal resolution, capturing the dynamics of relatively small and fast turbulent eddies (Churchfield et al., 2012; Chatelain et al., 2013; Yang et al., 2021; Li et al., 2022). While the range of resolved scales in LES is constrained by computational resources, the number of LES grid-points in typical simulations continues to increase. However, data handling and post-processing capabilities have not kept pace with the resulting rapid increase in data volumes. For instance, a single LES of turbulent flow outputting five field variables (e.g., the three velocity components, potential temperature and pressure) on 2,048<sup>3</sup> spatial grid points and integrated over, say, 10<sup>4</sup> time-steps (McWilliams et al., 1994; Alexakis et al., 2024), can generate Petabytes (PB) of data. As a result, most studies store only a few selected snapshots and rely heavily on pre-defined run-time diagnostics when time-resolved analysis is required. This approach reduces storage requirements but limits the ability to revisit data when new questions and concepts arise, often necessitating costly recomputation. Furthermore, certain analyses —such as backward-in-time particle tracking from an extreme dissipation event—cannot be performed without the full temporal data.

To address these challenges, modern database technologies have increasingly been applied to preserve and store data from simulation-based turbulence research (Perlman et al., 2007; Zhang et al., 2018; Chung et al., 2022; Duraisamy et al., 2019). One example is the Johns Hopkins Turbulence Database (JHTDB, https://turbulence.idies.jhu.edu), an open-access platform supported by the National Science Foundation (Perlman et al., 2007; Li et al., 2008). JHTDB enables researchers to interact with easily accessible, large-scale simulation data. The system currently hosts more than 1 PB of direct numerical simulation (DNS) data for canonical, turbulent flows of fundamental interest (over 2 PB if counting warm backup copies), including 6 space-time resolved datasets and several others with a few snapshots available. Some LES datasets of stably stratified atmospheric turbulence are also included in JHTDB. Through web-service-based tools, users can query the database using a "virtual sensors" interface, specifying spatial and temporal locations for which the system returns properly interpolated field or derivative values (Li et al., 2008; Yu et al., 2012). A hallmark of the platform is that it allows users to access only the specific subsets of the data they require, eliminating the need to download massive datasets or manage complex file formats. This approach has significantly broadened access to high-fidelity eddy-resolving simulation data and has contributed to democratizing high-performance computational turbulence research. To date, JHTDB data have been used in research reported in over 400 peer-reviewed journal articles.

At the same time, with the growing global demand for renewable energy, enhancing wind energy efficiency has become a key priority. As wind turbines grow larger and wind farms expand in scale, their interactions with the ABL become increasingly complex—particularly with respect to wake dynamics, energy extraction, and the redistribution of momentum within the flow. LES of large wind turbines have emerged as a crucial complement to field measurements, enabling researchers to explore flow-turbine interactions in detail and to develop engineering models that inform turbine placement strategies and improve wind farm efficiency. For example, Calaf et al. (2010) used LES with periodic boundary conditions to study the performance of "infinite" arrays of wind turbines under neutrally-stratified conditions. Abkar and Porté-Agel (2013, 2014) examined how wind farm density and free-atmosphere stability influence kinetic energy fluxes in a conventionally neutral boundary layer (CNBL) - defined as neutrally-stratified surface layers capped by stably-stratified free atmospheres (Zilitinkevich et al., 2002). Allaerts and Meyers (2015) explored the effect of capping inversion profile on wind farm performance. Numerous additional

LES-based studies have further advanced the field (Yang et al., 2014; Aitken et al., 2014; Martínez-Tossas et al., 2015; Stevens et al., 2018; Gharaati et al., 2022, 2024; Aiyer et al., 2024), highlighting the continued value of high-resolution simulation tools for understanding and optimizing wind energy systems.

These simulations, like many previous numerical studies of large-scale wind farms, generate extensive datasets. However, access to these data often remains restricted to the original researchers who conducted the simulations. The data (typically 4D space-time fields of velocity, temperature, etc.) are ephemeral: they must be analyzed in real-time during the simulation, or, at best, a limited number of snapshots are stored for post-processing, while the large majority of the data is discarded. As demonstrated in the case of the JHTDB database, providing access to the 4D space-time history of a simulation could provide substantial benefits for the broader research community. The value of open access to time-resolved numerical datasets is now being recognized beyond fluid dynamics, particularly in the fields of Geosciences. For example, the recently released NOW-23 dataset (Bodini et al., 2023) comprises a full year of Weather Research and Forecasting (WRF) model simulations of off-shore wind conditions over several expansive (100's km) U.S. coastal regions, offering valuable data for wind farm developers. However, no equivalent open-access LES datasets currently exist at smaller scales that explicitly include wind turbine effects—datasets that would be highly valuable for researchers focused on wake interactions, turbine siting, and wind farm optimization. More in general, the lack of data sharing in the wind energy sector has been recognized to hinder technical progress and leads to missed opportunities for improving the efficiency of energy markets (Kusiak, 2016)

To begin addressing the need for open access to LES wind farm data, we construct JHTDB-wind (see https://turbulence.idies.jhu.edu/datasets/windfarms, Zhu et al. (2025)), a publicly accessible turbulence database built on the JHTDB framework. This paper presents the dataset by detailing the simulation framework (Section 2), and flow configuration—specifically, a CNBL interacting with a 60-turbine wind farm using National Renewable Energy Laboratory (NREL) 5MW reference turbines. Here, CNBL is chosen because it is a less complicated atmospheric state, observed in nature (Liu and Stevens, 2022), for example, during the transition period after sunset or on cloudy days with powerful winds (Allaerts and Meyers, 2017; Liu et al., 2024). Simulation parameters are described in Section 3. The construction of the database system is described in Section 4, followed by an overview of representative data access methods based on the JHTDB virtual sensor method, illustrated here via Python examples (Section 5). Conclusions are summarized in Section 6. Further documentation is available directly on the database website.

#### 2 Large-eddy simulation framework

In this study, we use the open source LES code LESGO (https://lesgo.me.jhu.edu) as a numerical solver to simulate ABL flows and its interactions with wind turbines (Calaf et al., 2010; Stevens and Meneveau, 2017; Martinez et al., 2017; Stevens et al., 2018; Shapiro et al., 2018, 2020; Gharaati et al., 2022; Narasimhan et al., 2022, 2024a, 2025, 2024b; Gharaati et al., 2024; Ayala et al., 2024). The model represents all variables on a three-dimensional Cartesian grid, with x, y, and z denoting the streamwise, spanwise, and vertical directions, respectively. In index notation, these are expressed as  $x_i$  where i = 1, 2, 3. The corresponding velocities are denoted by  $u_i$ , or also with u, v, and w for its x, y, and z-direction components, respectively.

# 2.1 Governing equations and numerical methods

The turbulent flow is simulated by solving the filtered Navier-Stokes equations in their rotational form with Boussinesq thermal forcing and Coriolis effects, along with the transport equation for the potential temperature field. The governing equations include the filtered mass conservation,

$$\frac{\partial \tilde{u}_i}{\partial x_i} = 0,\tag{1}$$

the filtered momentum conservation,

$$\frac{\partial \tilde{u}_i}{\partial t} + \tilde{u}_j \left( \frac{\partial \tilde{u}_i}{\partial x_j} - \frac{\partial \tilde{u}_j}{\partial x_i} \right) = -\frac{\partial \tilde{p}^*}{\partial x_i} + \frac{g}{\theta_0} (\tilde{\theta} - \theta_0) \delta_{i3} - \frac{\partial \tau_{ij}^{SGS,d}}{\partial x_j} - f_i + f_c (\tilde{u}_2 - V_g) \delta_{i1} - f_c (\tilde{u}_1 - U_g) \delta_{i2}, \tag{2}$$

and the filtered heat conservation,

$$\frac{\partial \tilde{\theta}}{\partial t} + \tilde{u}_j \frac{\partial \tilde{\theta}}{\partial x_i} = -\frac{\partial \Pi_j}{\partial x_i}.$$
 (3)

Here, the tilde indicates filtering at the LES grid scale  $\tilde{\Delta} = \sqrt[3]{\Delta x \, \Delta y \, \Delta z}$ ;  $\rho$  is the density of air;  $\tau_{ij}^{\text{SGS}} = \widetilde{u_i u_j} - \widetilde{u}_i \widetilde{u}_j$  is the unresolved subgrid-scale (SGS) stress tensor, and  $\tau_{ij}^{\text{SGS},d} = \tau_{ij}^{\text{SGS}} - \delta_{ij} \tau_{kk}^{\text{SGS}}/3$  is the deviatoric (trace-free) part of  $\tau_{ij}^{\text{SGS}}$ , where  $\delta_{ij}$  is the Kronecker delta;  $\tilde{p}^* = \tilde{p}/\rho + \tilde{u}_k \tilde{u}_k/2 + \tau_{kk}^{\text{SGS}}/3$  is the pseudo pressure, where  $\tilde{p}$  is the resolved pressure;  $g = 9.81 \, \text{m/s}^2$  is the gravitational acceleration;  $\theta_0$  is the reference potential temperature scale; and  $f_i$  is the distributed body force for modeling the turbine-induced aerodynamic forces on the air flow (see §2.3). In the present study,  $\tau_{ij}^{\text{SGS},d}$  is parameterized using the Lilly-Smagorinsky eddy-viscosity type model (Smagorinsky, 1963; Lilly, 1966), i.e.,  $\tau_{ij}^{\text{SGS},d} = -2v_{\text{SGS}}\tilde{\delta}_{ij} = -2(C_s\tilde{\Delta})^2|\tilde{S}|\tilde{s}_{ij}$ , where  $\tilde{S}_{ij} = 0.5(\partial \tilde{u}_i/\partial x_j + \partial \tilde{u}_j/\partial x_i)$  is the resolved strain-rate tensor,  $|\tilde{S}| = \sqrt{2\tilde{s}_{ij}\tilde{s}_{ij}}$  is the strain-rate magnitude, and  $v_{\text{SGS}} = (C_s\tilde{\Delta})^2|\tilde{S}|$  is the modeled SGS eddy viscosity. The coefficient  $C_s$  is dynamically determined using the Lagrangian-averaged scale-dependent dynamic model (Bou-Zeid et al., 2005), which has been successfully applied in several prior LES studies of wind turbine wake flows (Calaf et al., 2010; Stevens and Meneveau, 2017; Martinez et al., 2017; Stevens et al., 2018; Narasimhan et al., 2022; Gharaati et al., 2022; Narasimhan et al., 2024a; Gharaati et al., 2024). In Eq. (3), the term  $\Pi_j = u_j \theta - \tilde{u}_j \theta$  is the SGS heat flux whose eddy diffusivity ( $\kappa_{\text{SGS}}$ ) is determined from  $\kappa_{\text{SGS}} = Pr_{\text{SGS}}^{-1}v_{\text{SGS}}$ , where the SGS Prandtl number of  $Pr_{\text{SGS}} = 1$  (Narasimhan et al., 2022) is prescribed.

The atmospheric boundary layer flow is driven by a geostrophic wind whose pressure gradient is given by  $-\nabla P_{\infty}/\rho = (f_c V_g, -f_c U_g)$ . Here,  $f_c = 2\Omega \sin \phi = 10^{-4} \, \mathrm{s}^{-1}$  is the Coriolis parameter corresponding to a mid-latitude position (specifically to  $\phi = 43.44^{\circ}$  with Earth's rotation rate  $\Omega = 7.27 \times 10^{-5} \, \mathrm{rad/s}$ ). The quantities  $U_g, V_g$  are the geostrophic wind velocity components along the x and y directions, respectively, with magnitude  $G = \sqrt{U_g^2 + V_g^2}$ , and directed at an angle of  $\alpha_G$  relative to the x direction such that  $U_g = G \cos \alpha_G$ ,  $V_g = G \sin \alpha_G$ . At each timestep, a proportional-integral (PI) controller is utilized to control the direction of the geostrophic wind such that the wind flows in the streamwise direction with zero wind veer at the hub height (Sescu and Meneveau, 2014; Narasimhan et al., 2022).

LESGO uses a Fourier-series-based pseudo-spectral method based on collocated grids for the spatial discretizations in the horizontal (x and y) directions, and a second-order central-difference method based on staggered grids in the vertical (z)

direction. The 3/2-rule is used to eliminate the aliasing error associated with the pseudo-spectral discretization of the nonlinear convective terms. The simulation is advanced in time using a fractional-step method. First, the velocity field is advanced in time by integrating Eq. (2) using the second-order Adams-Bashforth scheme to obtain a predicted velocity field. Then a pressure Poisson equation is constructed based on the divergence-free constraint Eq. (1) for the new time step and is solved to obtain the pseudo-pressure field. Lastly, the predicted velocity field is projected to the divergence-free space using the gradient of the pseudo pressure to obtain the velocity field for the new time step. The above fractional steps are repeated at every time step in LES to advance the flow field in time. More details of the numerical schemes used in the LESGO solver can be found in the original references (Albertson, 1996; Albertson and Parlange, 1999).

# 2.2 Boundary conditions

In the streamwise (x) direction, inflow–outflow boundary conditions are applied using the concurrent precursor simulation approach (Stevens et al., 2014). Specifically, a separate precursor domain without wind turbines is simulated to generate realistic turbulent inflow conditions, which are then imposed at the inlet of the wind farm domain. To ensure periodicity, a fringe region is introduced at the end of the wind farm domain where the outflow is gradually forced to match the inflow from the mapped region in the precursor domain. More details of the inflow-outflow conditions implemented in the current pseudo-spectral solver are provided in Stevens et al. (2014). Additionally, the simulation in the precursor domain uses a shifted periodic boundary condition where the flow field in a spanwise shifting region is shifted to prevent persistent spanwise locking of large-scale turbulent structures (Munters et al., 2016). Following the recommendation in Munters et al. (2016) a shift of  $L_{y-\text{shift}} = 0.25L_z$  is used in this study, where  $L_z$  is the domain height. In the spanwise (y) direction, periodic boundary conditions are used. In the vertical (z) direction, the ground surface boundary condition is specified in both the precursor and wind turbine domains using the Monin-Obukov Similarity Theory (MOST)-based equilibrium surface flux modeling (Monin and Obukhov, 1954). The components of local surface shear stress are computed as a function of the prescribed roughness length according to

$$\tau_{i,3|\text{surf}} = -u_*^2 \frac{\widehat{u_i}}{\sqrt{\widehat{u}^2 + \widehat{v}^2}}, \quad i = 1, 2; \quad \text{and} \quad u_* = \kappa \frac{\sqrt{\widehat{u}^2 (0.5\Delta z) + \widehat{v}^2 (0.5\Delta z)}}{\ln(0.5\Delta z/z_0)}. \tag{4}$$

Here,  $\kappa=0.41$  is the von Kármán constant,  $z_0$  is the prescribed roughness length, the friction velocity  $u_*$  is expressed in terms of the horizontal velocity  $(\widehat{u},\widehat{v})$  at the first grid-point  $(z_1=0.5\Delta z)$ , filtered at twice the grid resolution,  $\widehat{\Delta}=2\widetilde{\Delta}$  (Bou-Zeid et al., 2005). Since we simulate conventionally neutral conditions, the surface heat flux is set to zero, and thus no stability correction terms (as used in Xiao et al. (2025)) are included. At the top of the domain, a stress-free boundary condition is imposed. A sponge or Rayleigh-damping layer (Durran and Klemp, 1983) is included approaching the top boundary, ranging from  $0.75L_z$  to  $L_z$ , with a sponge inverse relaxation time-scale (frequency) parameter of  $3.9 \times 10^{-3} \, \mathrm{s}^{-1}$ . In this layer, a damping body force with a cosine profile is applied to suppress the reflection of gravity waves.

Henceforth, the  $(\tilde{\cdot})$  notation for LES-filtered field variables (e.g., velocity  $\tilde{u}_i$ , temperature  $\tilde{\theta}$ ) will be omitted for brevity. All subsequent variables should be interpreted as implicitly filtered quantities obtained from the LES solution, governed by the equations presented in this Section.

## 2.3 Wind turbine representation

The aerodynamic forces exerted by wind turbines on the airflow are modeled through the distributed body force term  $f_i$  in the momentum transport equations (Eq. (2)). During the initial spin-up phase (i.e, Phase 1), we employ an actuator disk model (ADM) on a coarse grid for computational efficiency, with the thrust force magnitude calculated as  $f = \frac{\pi}{8}\rho C_T'\langle u_T\rangle_d^2D^2$  (Calaf et al., 2010; Howland et al., 2016). Here,  $\rho$  is the air density,  $\langle u_T\rangle_d$  is the local wind velocity averaged over the rotor disk, D is the diameter of the wind turbine, and  $C_T'$  is the local thrust coefficient (set to a common value  $C_T' = 1.33$ ). We recall that  $C_T'$  is based on the disk-averaged velocity  $\langle u_T\rangle_d$  which, unlike the far-upstream velocity  $U_\infty$ , is immediately available in LES (Calaf et al., 2010).

After the spin-up simulation converges to quasi-steady behavior, the grid is refined to its final resolution, and the actuator line model (ALM) is adopted (Sørensen and Shen, 2002; Troldborg, 2009; Jha et al., 2014; Martínez-Tossas et al., 2015). In ALM, each turbine blade is represented by a collection of actuator points along a line, where forces are applied according to the velocity field and the angle of attack. The forces per unit width at every actuator point are computed as

$$\mathbf{f}_{\text{alm}} = 0.5\rho c |\mathbf{V}_{\text{rel}}|^2 (C_L \mathbf{e}_L + C_D \mathbf{e}_D), \tag{5}$$

where c is the airfoil chord length,  $|\mathbf{V}_{rel}|$  is the magnitude of the relative velocity of the upwind flow to the turbine blade,  $C_L$  and  $C_D$  are lift and drag coefficients obtained from tabulated airfoil data, and  $\mathbf{e}_L$  and  $\mathbf{e}_D$  are unit vectors along the direction of the lift and drag forces at each actuator point, respectively. These forces are then smeared using a Gaussian kernel to project them into the computational LES grid:

$$\eta_{\varepsilon} = \frac{1}{\varepsilon^3 \pi^{3/2}} e^{-r^2/\varepsilon^2},\tag{6}$$

where r is the distance from the grid point to the actuator point, and  $\varepsilon$  denotes the width of the kernel. The kernel width is chosen to be at least  $\varepsilon = 2(\Delta_x \Delta_y \Delta_z)^{1/3}$ , as recommended to avoid numerical instabilities (Troldborg, 2009; Martínez-Tossas et al., 2015).

The accuracy of the ALM can be sensitive to grid resolution and the choice of  $\varepsilon$ . The optimal  $\varepsilon_{opt}$  needed to resolve the induced velocities is typically much smaller than the  $\varepsilon$  used to avoid numerical instabilities (Martínez-Tossas et al., 2017). To address this challenge, we use the generalized filtered lifting line theory correction to accurately represent the blade aerodynamics (Martínez-Tossas and Meneveau, 2019; Martínez-Tossas et al., 2024), including the shedding of unresolved vorticity leading to missing induced velocities at the blade. The correction accounts for subgrid-scale induced velocity that would be obtained by using an optimal  $\varepsilon_{opt}$  by estimating its contribution and adding it to the resolved velocity in the LES. With the correction, the ALM provides consistent blade loading predictions across varying grid resolutions.

The NREL-5MW baseline wind turbine (Jonkman et al., 2009) is adopted as our reference model. It is a widely-used benchmark model developed by NREL to standardize research on wind technologies. The turbine has a diameter of  $D = 126 \,\mathrm{m}$ ,

three blades, and a hub height at elevation  $z_h = 90 \,\text{m}$ . It reaches a rated electrical power output of 5 MW at a rated wind speed of approximately 11.4 m/s. Its rotor blades utilize the DU (Delft University) and NACA (National Advisory Committee for Aeronautics) series airfoil profiles optimized for aerodynamic efficiency, structural integrity, and minimal fatigue loads, making the NREL-5MW turbine an essential tool for evaluating wind turbine performance, control strategies, structural design, and offshore platform dynamics.

The dataset employs fixed but row-dependent rotor angular velocities determined through an initialization procedure. Initialization begins with all turbines operating at tip-speed ratio TSR=7.5 (near-optimal for NREL-5MW turbines). In this initialization simulation (i.e., first part of Phase 2), the angular velocity  $\Omega$  for each turbine is then computed dynamically using:

$$\Omega = TSR \times \frac{1.087 \, U_d}{(1-a)R},\tag{7}$$

where  $U_d$  is the disk-averaged velocity; the numerator incorporates an empirical 8.7% correction factor for LES filter-scale effects ( $\varepsilon = 16 \,\mathrm{m}$ ), validated through single-turbine laminar inflow tests; the induction factor a derives from rotor geometry (blade number  $N_b = 3$ , radius  $R = 63 \,\mathrm{m}$ , and chord  $c = 3-4 \,\mathrm{m}$ ) and local inflow angle  $\phi$  via:

$$a = \frac{1}{(4\sin^2\phi)/(\sigma_r C_n) + 1},\tag{8}$$

with rotor solidity  $\sigma_r = N_b c/(\pi R)$  and force coefficient  $C_n = C_L \cos \phi + C_D \sin \phi$ . After approximately 40 minutes of initialization simulation, the angular velocity  $\Omega$  for each turbine is averaged within its respective row, which serves as the fixed operational values for the subsequent database simulations.

We also note that LESGO's ALM implementation includes detailed turbine operation control methods, such as pitching the blades (feathering) during region III operations, e.g., above rated conditions. In the current simulation we chose to operate all turbines exclusively at optimal tip-speed ratio, "region II" (also without including regions 1.5 and 2.5). This choice was made in order to avoid the need to store additional data relating to blade pitch (curtailment) and other complex turbine control actions. Since this practice deviates slightly from the reference NREL-5MW nameplate data, we refer to the turbine in our simulations as the NREL-5MW+ turbine. Indeed, the front turbines are allowed to rotate slightly faster than the maximum rotation rate of the original NREL-5MW reference turbine.

# 3 Simulation parameters

We simulate turbulent flow through a  $10 \times 6$  array of NREL-5MW+ turbines (with diameter  $D = 126 \,\mathrm{m}$ ) in a  $28.224 \times 3.78 \times 2 \,\mathrm{km}^3$  domain, equally split between precursor and wind farm subdomains (each  $112D = 14.112 \,\mathrm{km}$  long). Fig. 1 displays the domain dimensions. The precursor domain includes the region denoted as P of length  $5L_x/8$ , mapping region  $P_M$  of length  $L_x/8$ , and spanwise shifting region  $P_S$  of length  $L_x/8$ . The wind farm domain features 14D of upstream buffer zone, 63D turbine region, 21D downstream wake recovery region (these three regions combined are denoted as W), and 14D outflow fringe region ( $W_F$ ). The turbines are spaced 7D (streamwise) and 5D (spanwise), with lateral boundaries 2.5D from the outermost turbines.

Note that the fringe region  $W_F$ , as well as the mapping  $(P_M)$  and spanwise shifting  $(P_S)$  regions, have a length of  $L_x/8$ , and the mapping region  $P_M$  extends from  $5L_x/8$  to  $3L_x/4$ . Vertically, a 0.5 km Rayleigh damping sponge layer (denoted as R) is located between 1.5 and 2 km (see Figure 1). We adopt  $\theta_0 = 263.5$  K as the reference potential temperature, consistent with the value chosen in studies by Gadde and Stevens (2021) and our prior simulations of SBL and CNBL flows reported in Narasimhan et al. (2024a). This reference temperature was inspired by observations from the Beaufort Sea Arctic Stratus Experiment (BASE) and simulations by Kosović and Curry (2000). While the value of  $\theta_0$  is relatively low, it serves primarily as a relative additive reference that does not significantly affect the simulated flow dynamics or the physical interpretation of the results. For example, if we used 273K, it would change the implied thermal expansion coefficient in our Boussinesq approximation only by about 3%.

**Figure 1.** Schematic representation of the computational simulation domain (not to scale), showing: (a) top view (x–y plane), (b) side view (x–z plane) and (c) front view (y–z plane). The precursor computational domain consists of the regions denoted as "P", the precursor mapping region " $P_M$ ", and the precursor spanwise shifting region " $P_S$ ". The wind farm computational domain includes the wind farm region "W" and the fringe region " $W_F$ " near the outlet. Both precursor and windfarm computational domains include a Rayleigh damping region at the top (denoted as "R"). The turbine diameter D = 126 m and hub height  $z_h = 90$  m are also marked.

The turbulent flow is driven by a constant geostrophic wind speed  $G = 15 \,\mathrm{m/s}$  at  $\alpha_g \approx -22.5^\circ$  to the *x* direction, with the angle controlled by a PI controller ( $K_P = 10$ ,  $K_I = 0.5$ ) to align hub-height mean wind velocity with the *x*-axis in the conventionally neutral boundary layer (Sescu and Meneveau, 2014; Narasimhan et al., 2022). The surface has roughness length  $z_0 = 0.1 \,\mathrm{m}$  and reference potential temperature  $\theta_0 = 263.5 \,\mathrm{K}$ . Initial conditions set  $U_g = 15 \,\mathrm{m/s}$  (streamwise) and  $V_g = 0 \,\mathrm{m/s}$ 

(spanwise), perturbed by random noise, while potential temperature decreases from 265 K at the surface with a 1 K/km lapse rate, including random perturbations below 1 km.

The numerical simulation is conducted in three consecutive phases to ensure proper flow development and statistical convergence.

- Phase 1: Coarse-resolution ADM spin-up: A 10-hour simulation using the ADM is performed to establish a quasistationary atmospheric boundary layer and wind farm wake field. This phase leverages the computational efficiency of ADM, which approximates turbine forces without resolving actuator line-level aerodynamics.
- Phase 2: Fine-resolution ALM convergence. A 1-hour simulation using the actuator line model at finer spatial resolution transitions the flow from ADM-averaged to ALM-resolved turbine representation. Besides the turbine model update, two additional changes are introduced in this phase: (i) the time-stepping scheme is switched from a constant Courant–Friedrichs–Lewy (CFL) number of 0.0625 to a fixed time step of  $\Delta t = 0.025 \, \text{s}$ . This adjustment has negligible impact on the results because, under these simulation conditions, CFL = 0.0625 corresponds to  $\Delta t \approx 0.03 \, \text{s}$ . The slightly more restrictive  $\Delta t = 0.025 \, \text{s}$  maintains numerical stability while preserving solution accuracy. (ii) The rotor control changes from a fixed tip-speed ratio (TSR = 7.5) to fixed rotor angular velocities that vary across turbine rows, as tabulated in Table 1. This adjustment has a negligible impact on the results because the prescribed angular velocities closely match the values achieved under TSR = 7.5 conditions (see the calculation method in Section 2.3), ensuring nearly identical rotor dynamics.
- Phase 3: Fine-resolution simulation for database construction. A final 1-hour simulation is carried out to collect high-fidelity flow and turbine data. Flow field variables are recorded every 20 LES time steps (i.e., every 0.5s) on a filtered and subsampled spatial grid (every other grid point in the x-y plane), while wind turbine data—both integral and blade-resolved—are stored at every LES time step (0.025s). Note that we purposefully operate the NREL-5MW+ turbine in "region II" during the simulation time, in order to avoid having to choose and document additional controller actions. As a result, during some times some of the turbines operate "above rated conditions" but maintaining self-consistent aerodynamic behavior of the blades and air-flow.

**Table 1.** Rotor speed for each row of turbines.

| Row No.   | 1    | 2    | 3    | 4    | 5    | 6    | 7    | 8    | 9    | 10   |
|-----------|------|------|------|------|------|------|------|------|------|------|
| Ω (rad/s) | 1.33 | 1.02 | 1.04 | 1.07 | 1.09 | 1.09 | 1.09 | 1.09 | 1.09 | 1.10 |

The three phases of the simulation are illustrated through the time history of the boundary layer height  $z_i = h_{ABL}$  and the geostrophic wind angle shown in Fig. 2.

**Table 2.** Three consecutive phases and computational domain parameters

| Phase | Grid<br>level | Turbine<br>model | Domain size $(2 \times L_x) \times L_y \times L_z$                                | Number of grid points $(2 \times N_x) \times N_y \times N_z$ | Spatial resolution $\Delta x \times \Delta y \times \Delta z$ | Time grid CFL or $\Delta t$   |                             |  |
|-------|---------------|------------------|-----------------------------------------------------------------------------------|--------------------------------------------------------------|---------------------------------------------------------------|-------------------------------|-----------------------------|--|
|       |               |                  | $(km \times km \times km)$                                                        | ( 27 ) 2                                                     | $(m \times m \times m)$                                       | (- or s)                      |                             |  |
| 1     | Coarse        | ADM              | $(2\times14.112)\times3.78\times2$                                                | $(2 \times 512) \times 192 \times 400$                       | $27.56 \times 19.69 \times 5$                                 | CFL=0.0625                    |                             |  |
| 2     | Fine          | ALM              |                                                                                   |                                                              |                                                               | CFL=0.0625                    |                             |  |
|       |               | TSR=7.5          | (2 \ 14 112) \ 2 79 \ 2                                                           | (2 \ 1.526) \ \ 284 \ \ 400                                  | $9.19 \times 9.84 \times 5$                                   |                               |                             |  |
|       |               | ALM              | $(2 \times 14.112) \times 3.78 \times 2$                                          | $(2 \times 1,536) \times 384 \times 400$                     |                                                               | $\Delta t = 0.025 \mathrm{s}$ |                             |  |
|       |               | $\Omega = const$ |                                                                                   |                                                              |                                                               |                               |                             |  |
|       |               |                  |                                                                                   | Simulation with                                              |                                                               |                               |                             |  |
| 3     | Fine          | ALM              | $(2 \times 14.112) \times 3.78 \times 2$ $(2 \times 1,536) \times 384 \times 400$ |                                                              | $9.19 \times 9.84 \times 5$                                   | $\Delta t = 0.025 \mathrm{s}$ |                             |  |
|       |               | $\Omega = const$ | Sampling over/with                                                                |                                                              |                                                               |                               |                             |  |
|       |               |                  |                                                                                   | $(10.584 + 12.348) \times 3.78 \times 2$                     | $(576+672) \times 192 \times 400$                             | $18.38 \times 19.68 \times 5$ | $\Delta t = 0.5 \mathrm{s}$ |  |

**Figure 2.** Time history of boundary layer height  $z_i = h_{ABL}$ , and geostrophic wind angle  $\alpha$ , indicating the three simulation phases (Phase 1: Coarse-resolution ADM spin-up, Phase 2: Fine-resolution ALM convergence, and Phase 3: Fine-resolution simulation for database construction).

# 4 JHTDB-wind database construction

The LES data from the final 1-hour sampling period are systematically ingested into the database and organized into two primary data types: (i) flow field data, consisting of 4D space-time fields captured across both simulation domains (precursor and wind farm domains), providing complete spatiotemporal information about the atmospheric flow; and (ii) turbine data, which are further subdivided into two subtypes. The first subtype is turbine-level operational data, comprising time histories of turbine power and thrust. The second subtype is blade-level data, which includes time histories of aerodynamic quantities sampled at each discrete actuator point along each blade.

## 4.1 Flow field data

#### 4.1.1 Domain of the dataset

As described in Section 3, the LES is conducted in the domain of dimensions  $(2 \times 14.112) \times 3.78 \times 2 \,\mathrm{km}^3$  (see Table 2). When compiling the database, we exclude numerically imposed auxiliary regions: specifically, the final  $L_x/4$  of the precursor domain (which includes the precursor spanwise shifting region  $P_S$ ), and the final  $L_x/8$  of the wind farm domain (i.e., the wind farm fringe region  $W_F$ ), as visualized in Fig. 1. These regions serve purely numerical functions (periodicity enforcement and inflow recycling, respectively) without contributing to physical flow dynamics of interest. The resulting database domain has the extents of  $(10.584 + 12.348) \times 3.78 \times 2 \,\mathrm{km}^3$ , as shown in Fig. 3. The top 0.5km sponge region is kept in the database for simplicity of data management and possible interest.

Figure 3. Schematic representation of the database domain (not to scale). This is the physical domain available in the database, merging the precursor domain  $(P + P_M)$  up to the end of the mapping region at  $3/4L_x$ , with the windfarm domain (W) and excluding the fringe region  $(W_F)$ . A total of 60 turbines are shown, with only a subset labeled for clarity. The domain dimensions are  $(10.584 + 12.348) \times 3.78 \times 2 \text{ km}^3$ .

# 4.1.2 Spatial resolution of the dataset

To minimize storage, we applied spectral filtering on x-y planes for flow field data by truncating Fourier modes above  $\kappa_{\text{max}}/2$ , where  $\kappa_{\text{max}} = \pi/\Delta_{\text{LES}}$  is the LES cutoff wavenumber. The filtered fields were then subsampled at every alternate grid point in the x and y directions, maintaining the original vertical (z) resolution. This approach reduces the dataset size by 75% while maintaining fidelity in capturing the dynamically significant larger-scale flow structures and turbine wake interactions. Thus that the flow field data has a grid size of  $(576+672) \times 192 \times 400$ .

## 4.1.3 Temporal resolution of the dataset

Field data are stored at intervals of 0.5s (i.e., every 20 LES steps of 0.025s), ensuring that fluid parcels advected at the maximum geostrophic speed (15 m/s) travel less than the horizontal grid spacing ( $\Delta x \approx 9.19$  m) between snapshots. Although rotor blade tips move across several vertical grid spacings during this interval, the corresponding rotor force field is smooth (Gaussian filtered at scale  $\varepsilon = 16 m > 2\sqrt[3]{\Delta x \Delta y \Delta z}$ ), ensuring that the storage frequency of 0.5s remains appropriate. Over the 1-hour simulation period (i.e., 3,600 seconds), the simulation advances through 3,600/0.025=144,000 LES time steps, with flow fields stored at 144,000/20 = 7,200 consecutive snapshots.

#### 4.1.4 Final structure of the dataset

Consequently, the final stored data dimensions are  $n_x \times n_y \times n_z \times n_t = 1,248 \times 192 \times 400 \times 7,200$ . At each stored time step, six spatial fields are recorded: the three velocity components u(x,y,z,t), v(x,y,z,t), and w(x,y,z,t); the (kinematic) pressure field  $p(x,y,z,t)/p = p^*(x,y,z,t) - u_k u_k/2$  (the SGS stress trace is not available and is anyhow negligible); the potential temperature field relative to the reference temperature  $\theta'(x,y,z,t) = \theta(x,y,z,t) - \theta_0$ ; and the subgrid-scale eddy viscosity  $v_{\text{SGS}}(x,y,z,t)$ . In addition, the three components of the turbine force field,  $f_x(x,y,z,t)$ ,  $f_y(x,y,z,t)$ , and  $f_z(x,y,z,t)$ , are also stored. Unlike the other flow field variables, these force components are stored only from the ground up to 200m in the vertical direction. However, they are retained at the original spatial resolution (i.e., not filtered in the x planes). The detailed information of these stored field variables can be found in Table 3. It also needs to be mentioned that the concurrent precursor method ensures smooth transitions in velocity, potential temperature, and eddy viscosity fields between precursor and wind farm subdomains, by construction. However, due to the non-local nature of the pressure solution (solved separately in each domain via Poisson equations) and the velocity-only coupling between domains, the stored pressure field exhibits a minor discontinuity at the interface. This artifact does not affect the resolved turbulence dynamics or turbine wake interactions, but needs to be taken into account if computing pressure gradients across the boundary separating the precursor and wind farm domains.

These 4D field variables are stored using the Zarr format (Miles and et al., 2023). In Zarr-based storage, data are organized into chunks, the smallest units retrieved during a query. To ensure efficient data access, chunk sizes must be large enough to support common operations, such as differentiations and interpolations, that typically require access to a three-dimensional neighborhood around the query point, while remaining small enough to avoid excessive memory usage. Based on extensive testing and prior experience with other JHTDB datasets, a chunk size of 64<sup>3</sup> grid points provides optimal retrieval speeds

Table 3. Summary of flow field data variables.

| No. | Name of variable                     | Name in dataset | Symbol       | Unit              | Data size $n_x \times n_y \times n_z \times n_t$ | Data resolution $\Delta x \times \Delta y \times \Delta z \times \Delta t$ $(m \times m \times m \times s)$ |
|-----|--------------------------------------|-----------------|--------------|-------------------|--------------------------------------------------|-------------------------------------------------------------------------------------------------------------|
| 1   | Streamwise velocity                  |                 | и            |                   |                                                  |                                                                                                             |
| 2   | Spanwise velocity                    | velocity        | v            | m/s               |                                                  | $18.38 \times 19.68 \times 5 \times 0.5$                                                                    |
| 3   | Vertical velocity                    |                 | w            |                   | $1,248 \times 192 \times 400 \times 7,200$       |                                                                                                             |
| 4   | Potential temperature deviation      | temperature     | $\theta'$    | K                 | 1,246 × 192 × 400 × 7,200                        |                                                                                                             |
| 5   | Pressure (kinematic)                 | pressure        | p            | $m^2/s^2$         |                                                  |                                                                                                             |
| 6   | SGS eddy viscosity                   | eddyviscosity   | $v_{ m SGS}$ | m <sup>2</sup> /s |                                                  |                                                                                                             |
| 7   | Turbine streamwise force (kinematic) |                 | $f_x$        |                   |                                                  |                                                                                                             |
| 8   | Turbine spanwise force (kinematic)   | force           | $f_y$        | m/s <sup>2</sup>  | $871 \times 384 \times 40 \times 7{,}200$        | $9.19 \times 9.84 \times 5 \times 0.5$                                                                      |
| 9   | Turbine vertical force (kinematic)   |                 | $f_z$        |                   |                                                  |                                                                                                             |

and performance for typical data access modalities. We chose a similar chunk size but shaped according to  $52 \times 64 \times 80$  so that an integer multiple of the chunk size in each direction fits into the stored domain size. The total amount of data stored is about 15 Terabytes. These flow field data can be queried using getData(...) calls from analysis programs such as Python, MATLAB, Fortran, or C, in the same manner as with other turbulence datasets available through JHTDB.

## 4.2 Wind turbine data

#### 4.2.1 Turbine-level data

The turbine-level data are integral quantities characteristic of each turbine operation, which are derived from the actuator line modeling. This dataset includes high-fidelity time histories of power output, thrust force, and rotor angular velocity, sampled at  $\Delta t = 0.025 \,\mathrm{s}$  for all 60 turbines, as summarized in Table 4. In the present dataset, the angular velocity is held constant in time, but for other datasets (e.g., Xiao et al. (2025)), this is not generally the case. For each variable, the dataset consists of 144,000 rows and 2 columns, where the first column represents time and the second column contains the corresponding values of the recorded variable. The turbine data are stored in files using the Parquet format, which facilitates efficient access and querying from various programming languages. Turbine-level data can be accessed using the getTurbineData(...) function call from analysis environments such as Python or MATLAB.

**Table 4.** Summary of turbine-level data variables. Each dataset is a 2D matrix of size  $n_t \times 2$ , where  $n_t$  is the number of time steps. Columns 1 and 2 represent time and measured values, respectively.

| No. | Name of                | Name in  | Symbol | Unit  | Data size          | Data resolution |  |
|-----|------------------------|----------|--------|-------|--------------------|-----------------|--|
|     | variable               | dataset  |        |       | $n_t \times 2$     | $\Delta t$ (s)  |  |
| 1   | Power                  | power    | P      | W     |                    |                 |  |
| 2   | Thrust force           | thrust   | $F_t$  | N     | $144,000 \times 2$ | 0.025           |  |
| 3   | Rotor angular velocity | RotSpeed | Ω      | rad/s |                    |                 |  |

Table 4 summarizes the turbine-level data variables. Note that, unlike the field data which are stored in kinematic (density-independent) units, the force and power data require a specified air density. The value used in the simulations to compute these forces is  $\rho_{air} = 1.23 \text{ kg/m}^3$ .

#### 4.2.2 Blade-level data

In addition to the integral quantities characteristic of each turbine's operation, more detailed information is captured along each turbine blade to enable blade-resolved aerodynamic analysis. This fine-grained dataset allows users to investigate the local aerodynamic behavior of blades under unsteady flow conditions, which is critical for understanding load distributions, fatigue effects, and control optimization strategies. The turbine blade-level dataset includes high-fidelity time histories sampled at 0.025 s for all 180 blades in the wind farm (i.e., 60 turbines ×3 blades each), with aerodynamic and geometric quantities sampled at 100 discrete actuator line points along the blade span. As summarized in Table 5, a total of 19 variables are sampled and stored, with each variable written to a separate file. For each variable, the dataset has dimensions of 144,000×3 rows and 103 columns. Each time step includes three rows corresponding to the three blades of a turbine, resulting in a total of 144,000×3 rows. Vertically, the first column represents time in seconds, the second column specifies the turbine number, and the third column denotes the blade number (blades can be identified by the time-histories of the individual ALM point positions). The remaining 100 columns contain the values of the selected variables at each of the 100 actuator points from the blade root to tip. Similar as turbine-level data, blade-level data are stored as Parquet files, allowing efficient access across multiple programming environments. Blade-level data can be accessed using the getBladeData(...) function call from analysis environments such as Python or MATLAB.

**Table 5.** Summary of blade-level data variables. Each dataset is a 2D matrix of size  $(3 \times n_t) \times 103$ . Here,  $3 \times n_t$  represents the total number of blade-wise samples, formed by concatenating the time series data from each of the 3 blades of a turbine. Columns 1-3 represent time, turbine number, and columns 4-103 store aerodynamic measurements at  $n_{\ell} = 100$  discrete locations along each blade.

| No. | Name of variable                                                                       | Name in dataset | Symbol                | Unit   | Data size $(n_t \times 3) \times (n_\ell + 3)$ | Data resolution $\Delta t \times \Delta \ell$ (s × m) |
|-----|----------------------------------------------------------------------------------------|-----------------|-----------------------|--------|------------------------------------------------|-------------------------------------------------------|
| 1   | x-position of ALM point                                                                | xPos            | $P_X$                 |        |                                                |                                                       |
| 2   | y-position of ALM point                                                                | yPos            | $P_{y}$               | m      |                                                |                                                       |
| 3   | z-position of ALM point                                                                | zPos            | $P_z$                 |        |                                                |                                                       |
| 4   | Perturbation velocity at LES resolution, component 1                                   | uy_LES1         | $u'_{y, \text{LES}1}$ |        |                                                |                                                       |
| 5   | Perturbation velocity at LES resolution, component 2                                   | uy_LES2         | $u'_{y, \text{LES2}}$ |        |                                                |                                                       |
| 6   | Perturbation velocity at optimal resolution $(0.25c)$ , component 1                    | uy_opt1         | $u'_{y, \text{opt}}$  |        |                                                |                                                       |
| 7   | Perturbation velocity at optimal resolution $(0.25c)$ , component 2                    | uy_opt2         | $u'_{y,\mathrm{opt}}$ | m/s    |                                                |                                                       |
| 8   | Perturbation velocity correction $u'_{y,\text{opt}} - u'_{y,\text{LES}}$ , component 1 | du1             | $\Delta u'_{y,1}$     |        |                                                |                                                       |
| 9   | Perturbation velocity correction $u'_{y,opt} - u'_{y,LES}$ , component 2               | du2             | $\Delta u'_{y,2}$     |        | $(144,000 \times 3) \times (100 + 3)$          | $0.025 \times 0.615$                                  |
| 10  | Angle of attack                                                                        | alpha           | α                     | rad    |                                                |                                                       |
| 11  | Lift coefficient                                                                       | Cl              | $C_L$                 |        |                                                |                                                       |
| 12  | Drag coefficient                                                                       | Cd              | $C_D$                 | ] -    |                                                |                                                       |
| 13  | Lift force per unit length                                                             | lift            | $F_L/\ell$            | N/m    |                                                |                                                       |
| 14  | Drag force per unit length                                                             | drag            | $F_D/\ell$            | 18/111 |                                                |                                                       |
| 15  | Local relative velocity magnitude                                                      | Vmag            | $V_{mag}$             |        |                                                |                                                       |
| 16  | Axial component of the local relative velocity in blade-oriented coordinates           | Vaxial          | V <sub>axi</sub>      | m/s    |                                                |                                                       |
| 17  | Tangential component of the local relative velocity in blade-oriented coordinates      | Vtangential     | $V_{tan}$             |        |                                                |                                                       |
| 18  | Axial component of the local force                                                     | axialForce      | F <sub>axi</sub>      |        |                                                |                                                       |
| 19  | Tangential component of the local force                                                | tangentialForce | F <sub>tan</sub>      | N      |                                                |                                                       |

# 5 Web-accessible virtual sensor data access methods and examples

#### 5.1 Flow field data

A defining feature of the JHTDB database system (Li et al., 2008) is its low entry barrier for data usage, enabling users to efficiently explore large-scale simulation datasets through Web services and virtual sensor methodology. The JHTDB-wind system adopts the same approach, allowing access to wind farm data using these established tools. Users can develop analysis scripts or notebooks in familiar programming languages such as Python and Matlab (as well Fortran and C) to run them remotely on their own machines or on SciServer, a cloud service dedicated to running code close to the data. Within these analysis environments, users specify space-time arrays by defining spatial locations (e.g., along a line, across a surface, within a subvolume, or scattered arbitrarily) and corresponding time instances, i.e., users specify the positions of virtual sensor arrays. These space-time arrays are then passed to the predefined function, getData(...), which returns interpolated values of the selected variables at defined coordinates. This framework enables targeted, on-demand data access without the need to download large volumes of raw simulation output.

Figs. 4 and 5 display contour plots of flow field variables at the turbine hub height ( $z = z_h = 90 \,\text{m}$ ) for the precursor and wind farm domains, respectively.

Figure 4. Contour plots of instantaneous flow field variables in part of the precursor domain (here between x = 0 m and x = 10,381.875 m), at time t = 1,800.75 s. (a) the streamwise velocity u, (b) the vertical velocity w, (c) the pressure p, and (d) the potential temperature deviation  $\theta'$ .

Fig. 6 presents Python code snippets that demonstrate how to query the JHTDB-wind database to extract snapshots of velocity, pressure, and potential temperature fields at a specific time, approximately in the middle of the stored 1-hour dataset, namely at  $t = 1,800.75 \,\mathrm{s}$ . As a first step, an array "points" is populated with spatial coordinates that define a 2D plane: in this case, an equally spaced grid of  $950 \times 200$  points in the x and y directions at a constant height  $z = z_h = 90 \,\mathrm{m}$ . These query

points typically do not coincide with the actual simulation grid points, and users are not required to know the grid layout to access the data. The JHTDB-wind interface provides interpolated field values based on a user-specified interpolation method. Supported options include no interpolation (it returns the value at the nearest grid point), Lagrange Polynomials of order 4, 6, or 8, and several spline interpolation methods (Li et al., 2008; Graham et al., 2016). In this example, we use 8th-order Lagrange polynomial interpolation in space. Similarly, if the requested time does not coincide with a stored timestep, temporal interpolation is applied using third-order Piecewise Cubic Hermite Interpolating Polynomial (PCHIP) method (Li et al., 2008). This user-friendly data access model eliminates the need for downloading and parsing simulation files. Instead, the Python API (Application Programming Interface) returns arrays with the queried field variables, which can then be visualized directly within a Jupyter notebook (or Matlab code). This approach was used to generate Figs. 4 and 5. It is important to note that the full 1-hour dataset (comprising 14,400 timesteps) is available for analysis, allowing users to query any time between t = 0 and t = 3,600 s. For example, Fig. 7 shows a hub-height snapshot over the entire domain at time t = 2,505 s.

Figure 5. Contour plots of instantaneous flow field variables in part of the wind farm domain (here between x = 10,584 m and x = 21,921.375 m), at time t = 1,800.75 s. (a) the streamwise velocity u, (b) the vertical velocity w, (c) the pressure p, and (d) the potential temperature deviation  $\theta'$ . The short black lines represent the location of wind turbines.

Similar queries can be made for the values, spatial gradients, and Hessians (second-order derivatives) of all variables listed in Table 3. For example, Fig. 8(a) and (b) show turbine streamwise force-field  $f_x$  and the x-direction gradient of the pressure field  $(\partial p/\partial x)$ , respectively, on a y-z plane intersecting Row 1 (Turbines #1 - #6) at x=12,348 m (1764 m downstream of the wind farm domain), at time t=1000.013 s. Fig. 8(c) and (d) present similar results on a plane intersecting Row 9 (Turbines #49 - #54) at x=19,404 m (8,820 m downstream of the wind farm domain) at another time t=2,000.67 s. These plots were generated using the Python code shown in Fig. 9. In these examples, the queried times are intentionally chosen not to coincide with the stored simulation time steps, demonstrating the temporal interpolation capabilities of JHTDB-wind.

Next, we provide examples of computed mean vertical profiles of fundamental flow quantities within the precursor domain, which features standard conventionally neutral atmospheric conditions. Fig. 10 shows vertical profiles of horizontal- and time-

```
initialize getData parameters (except time and points)
initialize getData times and points, 'field'
initialize getData times and points
initialize getData times and points
initialize getData times and points
itime, nx, ny, n_points = 1800.75, 950, 200, 950 * 200

x_start, x_end, y_start, y_end = 10584, 21921.375, 0, 3789
x_points, y_points, z_points = np.linspace(x_start, x_end, nx, dtype=np.float64), np.linspace(y_start, y_end, ny, dtype=np.float64), points = np.array([[x, y, z_points] for x in x_points for y in y_points], dtype=np.float64)

use the tools and processing gizmos.
initialize getData(dataset, variable1, time, temporal_method, spatial_method, spatial_operator, points)
result1 = getData(dataset, variable2, time, temporal_method, spatial_operator, points)
result2 = getData(dataset, variable3, time, temporal_method, spatial_operator, points)
result3 = getData(dataset, variable3, time, temporal_method, spatial_operator, points)
result3 = getData(dataset, variable3, time, temporal_method, spatial_operator, points)
result3 = getData(dataset, variable3, time, temporal_method, spatial_operator, points)
```

**Figure 6.** Python code snippet used to obtain the data to generate the Fig. 5.

Figure 7. Contour plot of instantaneous streamwise velocity u in the entire database domain, ranging from x = 0 to x = 22,913.625 m, at time t = 2,505 s. It is noted that although the total length of the database domain is 10,584 + 12,348 = 22,932 m, the data resolution in x-direction is 18.375 m and the grid points are located at cell centers. Consequently, the last data point is located at 22,932 - 18.375 = 22,913.625 m. The short black lines represent the location of wind turbines.

averaged mean velocities, subgrid-scale eddy viscosity, and deviations in potential temperature, all obtained by averaging in the horizontal directions and over time. The data used to produce these profiles is retrieved using the virtual sensor framework, and an example code snippet demonstrating this process is shown in Fig. 11.

**Figure 8.** Instantaneous contours of turbine streamwise (i.e., x-component) force (as projected onto the LES grid using Gaussian smoothing as part of the ALM method) in y-z planes at (a) First row (i.e., Row 1, x = 12,348 m) and between the relevant vertical range  $z \in [2.5,200]$  m, and (c) second-to-last row (i.e., Row 9, x = 19,404 m). Panels (b) and (d) show the x-direction pressure gradient distributions on the same planes, coincident with the turbines.

```
initialize getData parameters (except time and points)

"""

variable], variable2, temporal_method, spatial_method1, spatial_method2, spatial_operator1, spatial_operator2 = 'force', 'pressure', 'none', 'none', 'fd4lag4', 'field', 'gradient'

"""

initialize getData times and points

"""

initialize getData times and points

"""

time1, time2, ny, nz, n_points = 1000.013, 2000.67, 384, 40, 384 * 40

y_start, y_end, z_start, z_end = 0, 3780, 2.5, 197.5

"""

first row of wind turbines locates 12348 m downstream of inlet.

second-to-last row of wind turbines locates 19404 m downstream of inlet.

"""

x_points1, y_points1, z_points1 = 12348, np.linspace(y_start, y_end, ny, dtype=np.float64), np.linspace(z_start, z_end, nz, dtype=np.float64)

x_points2, y_points2, z_points2 = 19404, np.linspace(y_start, y_end, ny, dtype=np.float64)

x_points2 = np.array([[x_points2, y, z] for y in y_points2 for z in z_points2], dtype=np.float64)

"""

use the tools and processing gizmos.

# process interpolation/differentiation of points.

result_force1 = getData(dataset, variable1, time1, temporal_method, spatial_method1, spatial_operator1, points2)

result_force2 = getData(dataset, variable2, time1, temporal_method, spatial_method2, spatial_operator2, points2)

result_pressure2 = getData(dataset, variable2, time2, temporal_method, spatial_method2, spatial_operator2, points2)
```

**Figure 9.** Python code snippet used to obtain the data to generate the Fig. 8.

**Figure 10.** Vertical profiles of horizontal- and time-averaged (a) velocities  $\langle u(z)\rangle_{x,y,t}$ ,  $\langle v(z)\rangle_{x,y,t}$  and velocity magnitude  $\mathcal{V}(z)_{x,y,t} = [\langle u(z)_{x,y,t}\rangle^2 + \langle v(z)_{x,y,t}\rangle^2]^{1/2}$ , (b, bottom axis) subgrid-scale eddy viscosity  $\langle v_{\text{SGS}}(z)\rangle_{x,y,t}$  used in the LES as a result of the Lagrangian scale-dependent dynamic model, (b, top axis) potential temperature deviation  $\langle \theta'(z)\rangle_{x,y,t}$  (i.e., the deviations from a reference temperature  $\theta_0 = 263.5 \, \text{K}$ ).

Figure 11. Python code snippet used to obtain the data to generate vertical profiles of  $\langle u(z)\rangle_{x,y,t}$ : for the 250 heights z between  $z=0.7\,\mathrm{m}$  and  $z=2,000\,\mathrm{m}$  separated by 8m, we query data on a regular mesh (not necessarily coinciding with stored grid points). For statistical convergence, we average over 4 times covering the entire hour (t=900;1,800;2,700;3,600) s.

## 5.2 Wind turbine data

Wind turbine data, including both the turbine-level and blade-level data, are considerably smaller than the 4D flow field data, and one possibility would have been to allow users to download these data directly as files. However, such an approach would

require users to identify specific files, understand naming conventions, and handle formatting, posing a barrier to seamless integration with flow field queries. To maintain consistency and usability across the platform, we adopt a similar virtual sensor data access paradigm used for the flow field data. Two dedicated query functions are developed: getTurbineData(...) for turbine-level quantities and getBladeData(...) for blade-resolved data. For getTurbineData(...), users specify the turbine number (ranging from 1 to 60) and desired time instances. For getBladeData(...), both turbine number and blade number (ranging from 1 to 3) need to be specified, along with an array of actuator point indices (ranging from 1 to 100) and times (ranging from 1 to 3600 s) at which the data are requested. Linear interpolation in time is supported to provide values between stored simulation time steps.

As an example, Fig. 12 presents the time series of total wind farm power output (Panel a) and of the Row 1 and Row 9 of six turbines (Panel b). The code snippet specifying the getTurbineData(...) call is shown in Fig. 13. Similar calls can be made to extract any of the turbine specific variables listed in Table 4.

**Figure 12.** Time evolution of power from turbines during the 10-minute time interval, i.e.,  $t \in [1000.33, 1600.33]$  s. (a) shows the total power from the entire wind farm, while (b) shows the power for the turbines in Row 1 (i.e., Turbines #1-#6) and in Row 9 (i.e., Turbines #49-#54).

Next, we illustrate the use of getBladeData(...) in Fig. 14, which shows (a) the time evolution of the lift and drag coefficients and (b) the lift coefficient as a function of blade angle. The blade angle is computed as  $\zeta(t) = \arctan[z(t) - z_h)/(x(t) - x_T)]$  over a 60-second period. The results shown are for a particular turbine and blade (Turbine #28 in the central portion of the wind farm and blade #3, the latter being an arbitrary choice, of course). The Python code snippet shown in Fig. 15 illustrates the use of getBladeData(...), with the queried data plotted directly as a time series within the same script. Using a similar approach, variable data can be extracted along turbine blades and further processed to compute higher-order statistics. Fig. 16 shows axial force, tangential force, drag and lift coefficients for an upstream turbine (Blade #1 of Turbine #1)

```
initialize getTurbineData parameters
"""

turbines = list(range(1, 61))
turbine_variable = 'power'

"""

initialize time array, below shows the one from 1000.33 s to 1600.33 s and the time interval is 0.025s
"""

time_start, time_end, dt = 1000.33, 1600.33, 0.025
ntime = int((time_end - time_start) / dt)
turbine_times = np.linspace(time_start, time_end, ntime, dtype=np.float64)
"""

use the tools and processing gizmos.
"""

# process turbine data.
turbine_result = getTurbineData(dataset, turbines, turbine_variable, turbine_times)
```

Figure 13. Python code snippet illustrating the use of the function getTurbineData(...) as part of a loop over all turbines in the wind farm, and subsequent summation to evaluate time-series of total power used to generate Fig. 12(a).

and a downstream turbine (Blade #1 of Turbine #60) at a specific time of t = 1,500 s. Any of the variables listed in Table 5 can be similarly queried (also in Matlab).

Figure 14. (a) Time evolution of lift and drag coefficients on an ALM point 80% along the span of Blade #3 for Turbine # 28. (b) Polar plot of lift coefficient for that point as a function of blade angle along its rotation. For this turbine, the rotational speed is fixed at  $\Omega = 1.09$  rad/s (as obtained from getTurbineData(...)), corresponding to approximately 10.5 revolutions during a 60-second period.

```
initialize getBladeData parameters
"""
turbines, blades, blade_variable1, blade_variable2 = [28], [3], 'Cl', 'Cd'
"""
initialize time array, below shows the one from 1000.33 s to 1600.33 s and the time interval is 0.025s
"""
time_start, time_end, dt = 1000.33, 1600.33, 0.025
ntime = int((time_end - time_start) / dt)
blade_times = np.linspace(time_start, time_end, ntime, dtype=np.float64)
blade_actuator_points = [80]
"""
use the tools and processing gizmos.
"""
# process blade data.
blade_result1 = getBladeData(dataset, turbines, blades, blade_variable1, blade_times, blade_actuator_points)
blade_result2 = getBladeData(dataset, turbines, blades, blade_variable2, blade_times, blade_actuator_points)
```

Figure 15. Python code snippet used to obtain the data to generate Fig. 14.

Figure 16. Distributions of ALM quantities along the turbine blade at a specific time (t = 1,500 s for Blade #1 of Turbine #1, blue lines; and Blade #1 of Turbine #60, orange lines: (a) Axial component of the local force (on each  $\Delta \ell = 0.615$  m segment)  $F_{axi}$ , (b) Tangential component of the local force (on each  $\Delta \ell = 0.615$  m segment)  $F_{tan}$ , (c) Lift coefficient  $C_l$ , (d) Drag coefficient  $C_d$ .

## 6 Conclusions

In this paper, we have introduced JHTDB-wind, hosting datasets from high-fidelity LES simulations of wind farms. We extend the standard "virtual sensors" data access methods (Li et al., 2008; Yu et al., 2012; Graham et al., 2016) that have been successfully used for democratizing access to more fundamental turbulence datasets. Besides velocity, pressure, potential temperature, and SGS eddy-viscosity fields, JHTDB-wind adds 4D space-time data on aerodynamic turbine force distributions as seen by the flow as well as time series of turbine and actuator line specific aerodynamic data along each of the turbine blades, modeled using ALM. We explain the simulation details and provide background on the numerical method and flow parameters, and provide detailed examples and explanations of the user-friendly data access methodologies. It is hoped that these data will provide useful insights about the complex fluid dynamic processes occurring in wind farms.

We realize that in generating a dataset for a representative conventionally neutral boundary layer case, with a relatively large wind farm with 60 turbines, many other choices could have been made (flow parameters, turbine model and control scheme, usage of a particular LES numerical code, numerical resolution, and so on). We anticipate that different members of the community would have made different choices, and we look forward to conversations about how to further improve such datasets. We believe, however, that the case selected is representative of CNBL wind farm dynamics that have been studied by many others before, with a well-tested numerical code. Hence, the authors hope that the data can be of some use and interest to researchers in wind energy.

As a final note, we have additionally prepared a second dataset for JHTDB-wind featuring an 8-turbine wind farm over a full diurnal cycle, capturing both strongly stable and unstable atmospheric boundary layer regimes at different times of the day and night (Xiao et al., 2025).

#### 7 Code and data availability

The wind farm data is available at the JHTDB-wind website at https://turbulence.idies.jhu.edu/datasets/windfarms (see also its DOI: https://doi.org/10.26144/D8ES-FC15). Various modes of data access are provided (Zhu et al., 2025): (i) Single-point queries of flow field variables using a browser interface at https://turbulence.idies.jhu.edu/database/query. (ii) Multiple point queries up to 4096 points at a time: downloading DEMO codes (Python or Matlab) at https://turbulence.idies.jhu.edu/database/wind and executing the DEMO code on user's own platforms. Users can then edit the DEMO codes to select different points and times to query desired data. To access current dataset, the "dataset" variable should be set to "nbl\_windfarm", with times chosen in the range 0–3600 seconds.

Author contributions. XZ performed the simulations, generated the majority of the data, and assisted in document and figure preparation, and detailed proof-reading. SX performed the majority of the data transformation into Zarr and Parquet formats, worked on testing data access methods, and generated many of the figures. GN developed the thermal stratification and initialization methods in the LES code. LAMT developed and implemented the generalized ALM method in the LES code. MS and HY developed the Giverny backend software

and Python/Matlab data access codes. GL directed the SciServer and zarr format optimization. AS designed the storage architecture. DG participated in data interpretation and analysis and manuscript editing. CM participated in simulation and database design, data interpretation and analysis, and document preparation and proof-reading.

Competing interests. We declare no competing interests.

Acknowledgements. The authors are grateful to IDIES staff for support in the creation of the JHTDB-wind database, as well as to Dr. Ned Patton for insightful comments regarding the simulations and data, and to Prof. Ben Schafer for his steadfast support and encouragement of JHTDB-wind. The project was made possible by a seed grant from the Ralph O'Connor Sustainable Energy Institute Research Initiative (ROSEI) at JHU, and by NSF grant #2034111 as well a joint NSF-DOE grant #2401013. The JHTDB project is supported by NSF (CSSI-2103874) and the Institute for Data Intensive Engineering and Science (IDIES) and its staff. We are grateful for the high-performance computing (HPC) resources and assistance received from both Cheyenne (doi:10.5065/D6RX99HX), made available by NCAR's CISL and sponsored by the NSF, and the Advanced Research Computing at Hopkins (ARCH) core facility (rockfish.jhu.edu), supported by the NSF under grant OAC1920103. This work was authored in part by NREL for the U.S. Department of Energy (DOE), operated under Contract No. DE-AC36-08GO28308. Funding provided by DOE Office of Energy Efficiency and Renewable Energy Wind Energy Technologies Office. The views expressed in the article do not necessarily represent the views of the DOE or the U.S. Government. The U.S. Government retains, and the publisher, by accepting the article for publication, acknowledges that the U.S. Government retains a nonexclusive, paid-up, irrevocable, worldwide license to publish or reproduce the published form of this work, or allow others to do so, for U.S. Government purposes.

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
