# Peer review of "JHTDB-wind: a web-accessible large-eddy simulation database of a wind farm with virtual sensor querying"

_Wind Energy Science, 2025_

## Referee Comment (RC1)

**Review: *JHTDB-wind: a web-accessible large-eddy simulation database of a wind farm with virtual sensor querying**

**General Comments**

This paper documents the model numerics and configuration for a database of LES of simulated wind farms in conventionally neutral conditions. The paper outlines the model that is used, how the simulations are set up, the assumptions that are made, and describes the output. Useful code is provided for querying the database. The simulation process appears valid and the dataset could prove to be very useful to the broad wind energy community.

I feel that this is a very well-written paper and clearly organized. I think this dataset is very helpful for the community and wish it luck in its acceptance and use for a large variety of applications. I appreciate the authors mentioning that other scientists within the community would have made other choices in the model setup and do believe that their setup is valid and justifiable. I have my reservations about the applicability of neutral stability but recognize the importance of simplicity for generalizable use.

I attempted to run the associated jupyter notebooks from the SciServer website but was unable to download the data (the command "getData" was initiated but did not progress). Thus, I was unable to verify the data on my end.

With all that said, **I recommend that the paper be accepted with only minor revisions / suggestions**.

**Major Revisions**

None

**Minor Revisions**

- The names, JHTDB and JHTDB-wind, are very close and line 39-40 had me thinking that the database being introduced had already been used in peer-reviewed publications over 400 times. While changing the name is probably too much to ask, clarifying in the sentence on 39-40 would probably be helpful to readers. Further, "JHTDB-DNS" is introduced on line 58. I imagine this is referring to the JHTDB dataset, but again - please make this clear to readers.
- I had missed it in the text and was searching for the boundary layer height of the simulation once it reached its quasi-steady state and eventually found the line where it references Figure 2. Two thoughts (feel free to disregard): PBL Height is often referred to as $z_i$ in meteorology and from what I have seen in wind energy research so you may

consider changing that here so it is more searchable, and justifying that the domain size and Rayleigh damping layer are sufficiently positioned. For example, I believe neutral conditions should have domain spans that are greater than $\sim 3*z_i$, and the damping layer should be just greater than $\sim 1.2*z_i$. Both of these conditions appear to be met, but I had my doubts and had to dig into the paper to find it. It may just be helpful to be upfront with it to justify that the model is set up well.

- Figure 4, 5 - why are the temperature deviations positive (2.0 += 0.02° K)? How are these calculated? I see this mentioned now in Figure 10's caption… this is probably not the place for defining how the field is calculated. Please move this up to where Figure 4 is being introduced.

**Technical Suggestions**

- The citations in the introduction have some issues (e.g., citation following a period on line 30, citation following a citation on line 31, etc.). Please carefully review the manuscript.

---

## Author Response (AR1)

**Responses to Referee 1, for WES submission "JHTDB-wind: a web-accessible large-eddy simulation database of a wind farm with virtual sensor querying"**

We are grateful to the reviewer for the positive appraisal of the paper, recommending publication with minor revisions and the useful comments. Regarding the associated jupyter notebook, we are sorry it did not work for the referee. We have not been able to replicate the problem on Sciserver and know for a fact that many users are already using and downloading data successfully using getData (on Sciserver but also using the Local notebook version than allow users to run the notebook on their own computer without needing an account on Sciserver). We are confident the system is working and is allowing users access to the data.

In the revision, we have addressed the referee's minor comments as follows:

- \* Minor revision comment 1: The names, JHTDB and JHTDB-wind, are very close and line 39-40 had me thinking that the database being introduced had already been used in peer-reviewed publications over 400 times. While changing the name is probably too much to ask, clarifying in the sentence on 39-40 would probably be helpful to readers. Further, "JHTDB-DNS" is introduced on line 58. I imagine this is referring to the JHTDB dataset, but again please make this clear to readers.
- Authors response: We agree with this comment and also that the additional naming "JHTDB-DNS" on line 58 caused confusion since even the existing database (without wind) included 2 LES datasets. We have changed the wording as follows, and do not use the naming JHTDB-DNS anywhere any longer. "JHTDB enables researchers to interact with easily accessible, large-scale simulation data. The system currently hosts more than 1 PB of DNS data for canonical, turbulent flows of fundamental interest (over 2 PB if counting warm backup copies), including 6 space-time resolved data sets and several others with a few snapshots available. Some LES datasets of stably stratified atmospheric turbulence are also included in JHTDB."
- \* Minor revision comment 2: I had missed it in the text and was searching for the boundary layer height of the simulation once it reached its quasi-steady state and eventually found the line where it references Figure 2. Two thoughts (feel free to disregard): PBL Height is often referred to as zi in meteorology and from what I have seen in wind energy research so you may consider changing that here so it is more searchable and justifying that the domain size and Rayleigh damping layer are sufficiently positioned. For example, I believe neutral conditions should have domain spans that are greater than ~3\*zi, and the damping layer should be just greater than ~1.2\*zi. Both of these conditions appear to be met, but I had my doubts and had to dig into the paper to find it. It may just be helpful to be upfront with it to justify that the model is set up well.
- Author response: Agreed. We have the boundary layer height height as z\_i in addition to h\_ABL everywhere, so readers searching for z\_i can find it easily. The domain height is 2z\_i and the sponge region is 0.5z\_i deep. For the purposes of the present dataset, we have checked that the effects of further increases of domain and sponge-layer height are negligible. For studies of larger-scale effects near the top of the boundary layer, e.g. creation and propagation of internal gravity waves generated by wind farms, indeed recent work has shown that domain heights as high as 10z\_i and higher would be requiredMany other LES have used domain heights 1.5z\_i and lower, etc. Our choice is a practical compromise considering data storage needs and costs.

- \* Minor revision comment 3: Figure 4, 5 why are the temperature deviations positive (2.0 += 0.02° K)? How are these calculated? I see this mentioned now in Figure 10's caption... this is probably not the place for defining how the field is calculated. Please move this up to where Figure 4 is being introduced.
- Author response: Thank you. Indeed, we missed defining what is theta-prime in the text (it is the deviation from the reference temperature theta\_0) and explaining that it is the variable stored in the database. We have corrected this oversight and now added both in the tables and on page 8: "We adopt  $\theta_0 = 263.5$  K as the reference potential temperature, consistent with the value chosen in studies by Gadde and Stevens (2021) and our prior simulations of SBL and CNBL flows reported in Narasimhan et al. (2024a). This reference temperature was inspired by observations from the Beaufort Sea Arctic Stratus Experiment (BASE) and simulations by Kosovic and Curry (2000). While the value of  $\theta_0$  is relatively low, it serves primarily as a relative additive reference that does not significantly affect the simulated flow dynamics or the physical interpretation of the results. For example, if we used 273K, it would change the implied thermal expansion coefficient in our Boussinesq approximation only by about 3%."
- \* Referee Technical Suggestion: The citations in the introduction have some issues (e.g., citation following a period on line 30, citation following a citation on line 31, etc.). Please carefully review the manuscript.
- Author response: Thank you for noticing these typos. They have been corrected in the revision.
- Author comment: in the revision, we also have corrected additional minor items and provided a more detailed explanation for our choice of running the turbine model in Region II only.

**Responses to Referee 2, for WES submission "JHTDB-wind: a web-accessible large-eddy simulation database of a wind farm with virtual sensor querying"**

We are grateful to the reviewer for the positive appraisal of the paper, recommending publication with minor revisions and the useful comments.

In the revision, we have addressed the referee's comments as follows:

- \* Point 1: Why was the reference temperature of 263.5K used rather than say the often-used value of 273K
- *Author response*: We have added the following clarification on page 8 regarding the choice of reference temperature:
- "We adopt  $\theta_0$  = 263.5 K as the reference potential temperature, consistent with the value chosen in studies by Gadde and Stevens (2021) and our prior simulations of SBL and CNBL flows reported in Narasimhan et al. (2024a). This reference temperature was inspired by observations from the Beaufort Sea Arctic Stratus Experiment (BASE) and simulations by Kosovic and Curry (2000). While the value of  $\theta_0$  is relatively low, it serves primarily as a relative additive reference that does not significantly affect the simulated flow dynamics or the physical interpretation of the results. For example, if we used 273K, it would change the implied thermal expansion coefficient in our Boussinesq approximation only by about 3%."
- \* Point 2: Typos: Line 200 delete the 'a' after '0.5 km', line 289 insert 'for' after '0.025 s'
- Author response: Thank you for noticing these typos. They have been corrected in the revision.
- Author comment: in the revision, we also have corrected additional minor items and provided a more detailed explanation for our choice of running the turbine model in Region II only.